# Estimating Total Correlation with Mutual Information Bounds

**Pengyu Cheng, Weituo Hao, Lawrence Carin**
Department of Electrical and Computer Engineering
Duke University
pengyu.cheng@duke.edu

## Abstract

Total correlation (TC) is a fundamental concept in information theory to measure the statistical dependency of multiple random variables. Recently, TC has shown effectiveness as a regularizer in many machine learning tasks when minimizing/maximizing the correlation among random variables is required. However, to obtain precise TC values is challenging, especially when the closed-form distributions of variables are unknown. In this paper, we introduced several sample-based variational TC estimators. Specifically, we connect the TC with mutual information (MI) and constructed two calculation paths to decompose TC into MI terms. In our experiments, we estimated the true TC values with the proposed estimators in different simulation scenarios and analyzed the properties of the TC estimators.

## 1 Introduction

Statistical dependency measures the correlation of random variables or factors in models, which is often an important concern in various scientific domains including statistics [12, 15], robotics [16, 4], bioinformatics [18, 24], and machine learning [7, 1, 14]. In recent deep learning studies, statistical dependency has increasingly served as learning objectives or regularizers for neural network training, and has achieved improvement in terms of model robustness [25], generalizability [1], interpretablity [7, 9], *etc.*

Among statistical dependency measurements, mutual information (MI) is commonly used in machine learning. Given two random variables $\boldsymbol{x}, \boldsymbol{y}$, the mutual information is defined as:

$$\mathcal{I}(\boldsymbol{x}; \boldsymbol{y}) = \mathbb{E}_{p(\boldsymbol{x},\boldsymbol{y})}\Big[ \log \frac{p(\boldsymbol{x},\boldsymbol{y})}{p(\boldsymbol{x})p(\boldsymbol{y})}\Big]. \tag{1}$$

Recently, mutual information has shown significant improvement when applied as a training criterion on learning tasks, such as conditional generation [7], domain adaptation [11], representation learning [6], and fairness [23]. However, MI can only handle the statistical dependency between two variables. When considering optimization of correlation among multiple variables, MI requires computation of each variable pair, which leads to a quadratic increase in computation cost. To address this problem, total correlation (TC) has been proposed by extending MI to multi-variable cases:

$$\mathcal{TC}(\boldsymbol{X}) = \mathcal{TC}(\boldsymbol{x}_1, \boldsymbol{x}_2, \dots, \boldsymbol{x}_n) = \mathbb{E}_{p(\boldsymbol{x}_1,\boldsymbol{x}_2,\dots,\boldsymbol{x}_n)}\Big[ \log \frac{p(\boldsymbol{x}_1, \boldsymbol{x}_2, \dots, \boldsymbol{x}_n)}{p(\boldsymbol{x}_1)p(\boldsymbol{x}_2)\dots p(\boldsymbol{x}_n)}\Big]. \tag{2}$$

TC has also proven effective to enhance machine learning models in many tasks, such as independent component analysis [3], and disentangled representation learning [5, 19, 17]. However, TC suffers from the same numerical problem as MI: the exact values of TC are difficult to calculate without the closed-form distribution $p(\boldsymbol{x}_i)$ and with only samples accessible. Previous works on disentangled representation learning [5, 10] avoid the estimation problem by assuming that both the latent priors and the inference posteriors follow multi-variate Gaussian distributions. Poole, *et al.* [22] proposed an upper bound of TC by further introducing another variable $\boldsymbol{y}$. With a strong assumption that given

34th Conference on Neural Information Processing Systems (NeurIPS 2020), Vancouver, Canada.

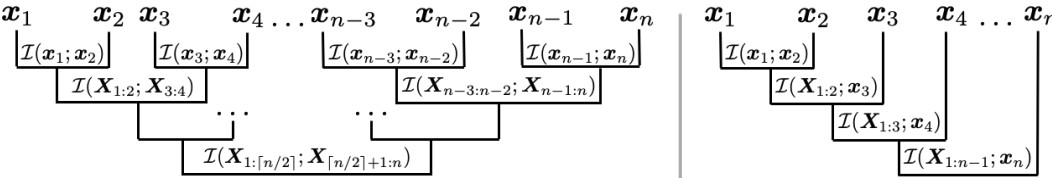

Figure 1: Two calculation paths of total correlation. **Left** (Tree-like calculation path): Divide the current variables into two subgroups with similar sizes. Calculate the MI between the subgroups and recursively calculate TC of both subgroups. $\lceil n/2 \rceil$ is the smallest number larger than $n/2$. **Right** (Line-like calculation path): Calculate the MI between the current group of variables and the next variable, and then add the the next variable into current group.

$\boldsymbol{y}$, all $\boldsymbol{x}_i|\boldsymbol{y}$ are independent, $p(\boldsymbol{X}|\boldsymbol{y}) = \prod_{i=1}^{n} p(\boldsymbol{x}_i|\boldsymbol{y})$, Poole, *et al.* [22] concluded that $\mathcal{TC}(\boldsymbol{X}) = \sum_{i=1}^{n} \mathcal{I}(\boldsymbol{x}_i; \boldsymbol{y}) - \mathcal{I}(\boldsymbol{X}; \boldsymbol{y})$. All the aforementioned methods require additional assumptions to the distributions, which limits their application scenarios.

In this paper, we propose two TC estimation strategies based on mutual information variational bounds. More specifically, we decompose TC into the summation of MI terms along two different calculation paths: the tree-like path and the line-like path. Then the TC values are approximated by applying MI estimation to each decomposed term. In our experiments, we test the performance of the proposed TC estimators under multivariate Gaussian simulations.

## 2 Method

With the definition of total correlation (TC) and mutual information (MI) in (2) and (1), we find a connection between TC and MI summarized in Theorem 2.1.

**Theorem 2.1.** *Suppose $\mathcal{A} = \{i_1, i_2, \ldots, i_m\} \subseteq \{1, 2, \ldots, n\}$ is an index subset. $\bar{\mathcal{A}} = \{j : j \notin \mathcal{A}\}$ is the complementary set of $\mathcal{A}$. Denote $\boldsymbol{X}_{\mathcal{A}} = (\boldsymbol{x}_{i_1}, \boldsymbol{x}_{i_2}, \ldots, \boldsymbol{x}_{i_m})$ as the selected variables from $\boldsymbol{X}$ with the indexes $\mathcal{A}$. Then we have $\mathcal{TC}(\boldsymbol{X}) = \mathcal{TC}(\boldsymbol{X}_{\mathcal{A}}) + \mathcal{TC}(\boldsymbol{X}_{\bar{\mathcal{A}}}) + \mathcal{I}(\boldsymbol{X}_{\mathcal{A}}; \boldsymbol{X}_{\bar{\mathcal{A}}})$.*

**Corollary 2.1.1.** *Given a variable group $\boldsymbol{X}$ and another $\boldsymbol{y}$, $\mathcal{TC}(\boldsymbol{X} \cup \{\boldsymbol{y}\}) = \mathcal{TC}(\boldsymbol{X}) + \mathcal{I}(\boldsymbol{X}; \boldsymbol{y})$.*

**Corollary 2.1.2.** *Given $\boldsymbol{X} = (\boldsymbol{x}_1, \boldsymbol{x}_2, \ldots, \boldsymbol{x}_n)$, we have $\mathcal{TC}(\boldsymbol{X}) = \sum_{i=1}^{n-1} \mathcal{I}(\boldsymbol{X}_{1:i}; \boldsymbol{x}_{i+1})$.*

The Theorem 2.1 provides insight that the TC of a group of variables $\boldsymbol{X}$ can be decomposed into the TC of two subgroups $\boldsymbol{X}_{\mathcal{A}}$ and $\boldsymbol{X}_{\bar{\mathcal{A}}}$ and the MI between the two subgroups. Therefore, we can recursively represent the TC with MI terms. More specifically, we propose two schemes with different structures to calculate TC with different MI terms (as shown in Figure 1).

Let $\boldsymbol{X}_{i:j} = (\boldsymbol{x}_i, \boldsymbol{x}_{i+1}, \ldots, \boldsymbol{x}_j)$ denote a subset of variables with indexes from $i$ to $j$. Based on Theorem 2.1, we propose two recursive TC calculation schemes: (1) **Line-like**: $\mathcal{TC}(\boldsymbol{X}_{1:i+1}) = \mathcal{TC}(\boldsymbol{X}_{1:i}) + \mathcal{I}(\boldsymbol{X}_{1:i}; \boldsymbol{x}_{i+1})$; (2) **Tree-like**: $\mathcal{TC}(\boldsymbol{X}_{i:j}) = \mathcal{TC}(\boldsymbol{X}_{i:\lfloor(i+j)/2\rfloor}) + \mathcal{TC}(\boldsymbol{X}_{\lfloor(i+j)/2\rfloor+1:j}) + \mathcal{I}(\boldsymbol{X}_{i:\lfloor(i+j)/2\rfloor}; \boldsymbol{X}_{\lfloor(i+j)/2\rfloor+1:j})$, where $\lfloor t \rfloor$ indicates the largest integer smaller than $t$. The line-like dynamic calculates the MI between a subgroup and a single variable, which leads to the representation of TC as the summation in Corollary 2.1.2. The tree-like dynamic divides the variables into balanced subgroups, so that the MI between two subgroups can be calculated with two variable parts in similar dimensions. Since the tree-like estimation is hard to summarize in an equation, we describe it in Algorithm 1. With the total correlation being decomposed into summation of MI terms, we can derive total correlation estimators based on the previous mutual information variational bounds.

---

**Algorithm 1** Tree-like TC estimation algorithm

---

    **Prerequisite:** MI estimation method $\hat{\mathcal{I}}$, samples $\{\boldsymbol{X}^{(i)}\}_{i=1}^{M} = \{(\boldsymbol{x}_1^{(i)}, \boldsymbol{x}_2^{(i)}, \ldots, \boldsymbol{x}_n^{(i)})\}_{i=1}^{M}$
    **Function** TC$_{\text{Tree}}$-estimate($\boldsymbol{X}_{i:j}$)**:**
    **if** $j - i \leq 0$ **then**
        **return** $0$
    **else**
        $m = \lfloor (i+j)/2 \rfloor$
        **return** TC$_{\text{Tree}}$-estimate($\boldsymbol{X}_{i:m}$) + TC$_{\text{Tree}}$-estimate($\boldsymbol{X}_{m+1:j}$) + $\hat{\mathcal{I}}(\boldsymbol{X}_{i:m}; \boldsymbol{X}_{m+1:j})$
    **end if**

---

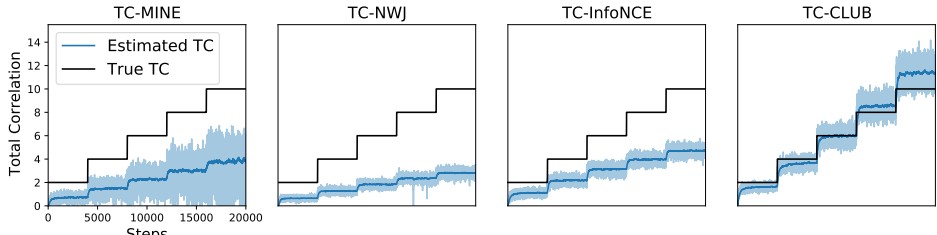

Figure 2: Simulation performance of TC **Line-like** estimators with different MI bounds.

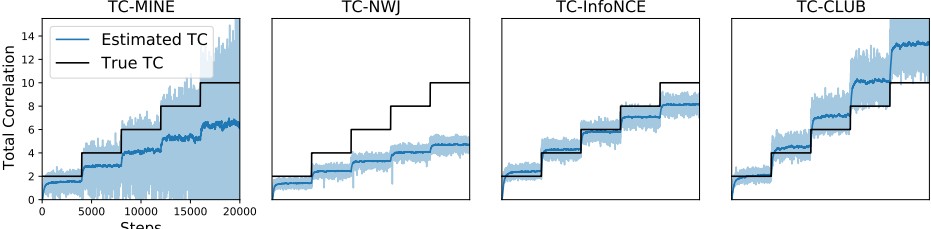

Figure 3: Simulation performance of TC **Tree-like** estimators with different MI bounds.

## 3 Experiments

We derive our TC estimators based on four MI bounds (MINE [2], NWJ [20], InfoNCE [21], and CLUB [8]) as TC-MINE, TC-NWJ, TC-InfoNCE, and TC-CLUB. The detailed description and implementation to the four MI estimators are shown in the Supplementary Material. Then we test the TC estimators with both tree-like and line-like strategies on simulations. The simulation data are drawn from four-dimensional Gaussian distributions $(\boldsymbol{x}_1, \boldsymbol{x}_2, \boldsymbol{x}_3, \boldsymbol{x}_4) \sim \mathcal{N}(\boldsymbol{0}, \boldsymbol{\Sigma})$, where $\boldsymbol{\Sigma}$ is a covariance matrix with all diagonal elements equal to $1$. With this Gaussian assumption, the true TC value can be calculated as $\mathcal{TC}(\boldsymbol{x}_1, \boldsymbol{x}_2, \boldsymbol{x}_3, \boldsymbol{x}_4) = -\frac{1}{2} \log \text{Det}(\boldsymbol{\Sigma})$, where $\text{Det}(\boldsymbol{\Sigma})$ is the determinant of $\boldsymbol{\Sigma}$. Therefore, we can adjust the correlation coefficients in $\boldsymbol{\Sigma}$ to set the ground-truth TC values in the range $\{2.0, 4.0, 6.0, 8.0, 10.0\}$. At each TC true value, we sample data batches 4000 times, with batch size equal to 64, for the training of variational TC estimators.

In Figure 2, we report the performance of our TC estimators with different MI bounds at each training steps. In each figure, the true TC value is shown as a step function with black line. The estimation values are displayed among different steps with shadow blue curves. The dark blue curves shows the local averages of estimated TC, with a bandwidth equal to 200. Under both a tree-like and line-like path calculation, the TC-MINE, TC-NWJ and TC-InfoNCE remains a lower bound of the truth TC values, based on the fact that MINE, NWJ, and InfoNCE are lower bound of mutual information. CLUB is an MI upper bound, while the TC-CLUB also behaves as an upper bound of total correlation.

The upper bound method TC-CLUB achieves better performance with line-like calculation. This is because that CLUB requires a variational approximation $q_\theta(\boldsymbol{v}|\boldsymbol{u})$ when estimating $\mathcal{I}(\boldsymbol{v}; \boldsymbol{u})$. When we use line-like calculation path, $\boldsymbol{v} = \boldsymbol{x}_{i+1}$ is always a single variable, and $\boldsymbol{u} = \boldsymbol{X}_{1:i}$ is the concatenation of $(\boldsymbol{x}_1, \ldots, \boldsymbol{x}_i)$. The $q_\theta(\boldsymbol{v}|\boldsymbol{u})$ as a neural network can have better performance with output $\boldsymbol{v}$ in a fixed low dimension. In contrast, the lower bound methods show better estimation with tree-like calculation than line-like calculation. Because for all listed lower bound methods, the estimation of $\mathcal{I}(\boldsymbol{v}; \boldsymbol{u})$ is based on $\boldsymbol{v}$ and $\boldsymbol{u}$ equally. With the tree-like strategy, each time the MI estimators are provided with samples in similar dimensions, which facilitates the learning of lower bound MI estimators. The bias and variance of the TC estimators are shown in the Supplementary Material.

## 4 Discussion

We have derived the line-like and tree-like calculation strategies to decompose the total correlation into summation of mutual information. By estimating mutual information terms with MI bounds, we introduced several TC estimators. The tree-like and line-like calculation strategies can bring advantages to TC estimation depending on different MI estimation processes. The proposed TC estimators can be further applied as learning criterion on many deep learning tasks, such as disentangled representation learning, ensemble learning, and model distillation.

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

# A  Proofs

*Proof of Theorem 2.1.* Note that $\boldsymbol{X}_{\mathcal{A}} := (\boldsymbol{x}_{i_1}, \boldsymbol{x}_{i_2}, \ldots, \boldsymbol{x}_{i_m})$ and $\boldsymbol{X}_{\hat{\mathcal{A}}} = \boldsymbol{X}/\boldsymbol{X}_{\mathcal{A}}$. Denote $\boldsymbol{X}_{\hat{\mathcal{A}}} = (\boldsymbol{x}_{j_1}, \boldsymbol{x}_{j_2}, \ldots, \boldsymbol{x}_{j_l})$. Then

$$
\begin{aligned}
\mathcal{TC}(\boldsymbol{X}) &= \mathbb{E}_{p(\boldsymbol{X})}\left[\log \frac{p(\boldsymbol{x}_1, \boldsymbol{x}_2, \ldots, \boldsymbol{x}_n)}{p(\boldsymbol{x}_1)p(\boldsymbol{x}_2)\ldots p(\boldsymbol{x}_n)}\right] \\
&= \mathbb{E}_{p(\boldsymbol{X})}\left[\log \left(\frac{p(\boldsymbol{X}_{\mathcal{A}})}{p(\boldsymbol{x}_{i_1})p(\boldsymbol{x}_{i_2})\ldots p(\boldsymbol{x}_{i_m})} \cdot \frac{p(\boldsymbol{X}_{\hat{\mathcal{A}}})}{p(\boldsymbol{x}_{j_1})p(\boldsymbol{x}_{j_2})\ldots p(\boldsymbol{x}_{j_l})} \cdot \frac{p(\boldsymbol{X})}{p(\boldsymbol{X}_{\mathcal{A}})p(\boldsymbol{X}_{\hat{\mathcal{A}}})}\right)\right] \\
&= \mathcal{TC}(\boldsymbol{X}_{\mathcal{A}}) + \mathcal{TC}(\boldsymbol{X}_{\hat{\mathcal{A}}}) + \mathcal{I}(\boldsymbol{X}_{\mathcal{A}}; \boldsymbol{X}_{\hat{\mathcal{A}}})
\end{aligned}
$$

$\square$

*Proof of Corollary 2.1.2.* We denote $\boldsymbol{X}_{i:j} := (\boldsymbol{x}_i, \boldsymbol{x}_{i+1}, \ldots, \boldsymbol{x}_{j-1}, \boldsymbol{x}_j)$. Note that

$$
\begin{aligned}
\mathcal{TC}(\boldsymbol{X}_{1:n}) &= \mathbb{E}_{p(\boldsymbol{x}_1, \boldsymbol{x}_2, \ldots, \boldsymbol{x}_n)}\left[\log \frac{p(\boldsymbol{x}_1, \boldsymbol{x}_2, \ldots, \boldsymbol{x}_n)}{p(\boldsymbol{x}_1)p(\boldsymbol{x}_2)\ldots p(\boldsymbol{x}_n)}\right] \\
&= \mathbb{E}_{p(\boldsymbol{x}_1, \boldsymbol{x}_2, \ldots, \boldsymbol{x}_n)}\left[\log \left(\frac{p(\boldsymbol{x}_1, \boldsymbol{x}_2, \ldots, \boldsymbol{x}_{n-1}, \boldsymbol{x}_n)}{p(\boldsymbol{x}_1, \boldsymbol{x}_2, \ldots, \boldsymbol{x}_{n-1})p(\boldsymbol{x}_n)} \cdot \frac{p(\boldsymbol{x}_1, \boldsymbol{x}_2, \ldots, \boldsymbol{x}_{n-1})}{p(\boldsymbol{x}_1)p(\boldsymbol{x}_2)\ldots p(\boldsymbol{x}_{n-1})}\right)\right] \\
&= \mathcal{I}(\boldsymbol{x}_1, \boldsymbol{x}_2, \ldots, \boldsymbol{x}_{n-1}; \boldsymbol{x}_n) + TC(\boldsymbol{X}_{1:n-1}) \\
&= \mathcal{I}(\boldsymbol{X}_{1:n-1}; \boldsymbol{x}_n) + \mathcal{TC}(\boldsymbol{X}_{1:n-1})
\end{aligned}
$$

Similarly,

$$
\mathcal{TC}(\boldsymbol{X}_{1:n}) = \mathcal{I}(\boldsymbol{X}_{1:n-1}; \boldsymbol{x}_n) + \mathcal{I}(\boldsymbol{X}_{1:n-2}; \boldsymbol{x}_{n-1}) + \mathcal{TC}(\boldsymbol{X}_{1:n-2}) = \sum_{i=1}^{n-1} \mathcal{I}(\boldsymbol{X}_{1:i}; \boldsymbol{x}_{i+1}) \tag{3}
$$

$\square$

# B  Experiment Results

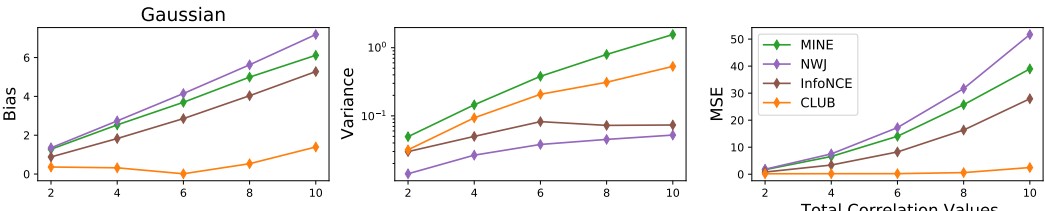

Figure 4: Bias, variance and MSE of **line-like** TC estimators

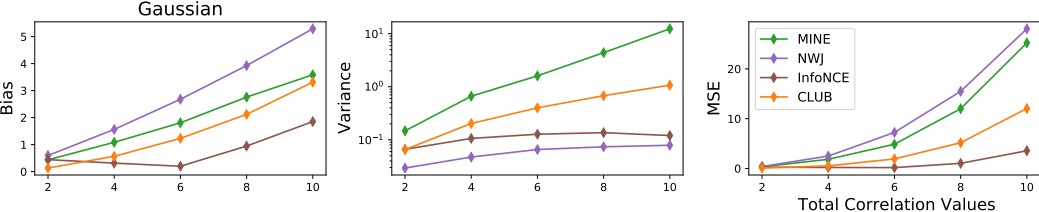

Figure 5: Bias, variance and MSE of **tree-like** TC estimators

## C MI Estimators

The Mutual Information Neural Estimator (MINE) [2] is defined as

$$\mathcal{I}_{\text{MINE}} := \mathbb{E}_{p(\boldsymbol{x},\boldsymbol{y})}[f(\boldsymbol{x},\boldsymbol{y})] - \log(\mathbb{E}_{p(\boldsymbol{x})p(\boldsymbol{y})}[e^{f(\boldsymbol{x},\boldsymbol{y})}]), \tag{4}$$

where $f(\cdot,\cdot)$ is a value function (or, a critic) approximated by a neural network.

The NWJ [20] lower bound is based on the $f$-divergence representation of MI:

$$\mathcal{I}_{\text{NWJ}} := \mathbb{E}_{p(\boldsymbol{x},\boldsymbol{y})}[f(\boldsymbol{x},\boldsymbol{y})] - \mathbb{E}_{p(\boldsymbol{x})p(\boldsymbol{y})}[e^{f(\boldsymbol{x},\boldsymbol{y})-1}]. \tag{5}$$

The InfoNCE [21] lower bound is based on Noise Contrastive Estimation (NCE) [13]:

$$\mathcal{I}_{\text{NCE}} := \mathbb{E}\left[\frac{1}{N}\sum_{i=1}^{N}\log\frac{e^{f(\boldsymbol{x}_i,\boldsymbol{y}_i)}}{\frac{1}{N}\sum_{j=1}^{N}e^{f(\boldsymbol{x}_i,\boldsymbol{y}_j)}}\right], \tag{6}$$

where the expectation is over $N$ samples $\{\boldsymbol{x}_i,\boldsymbol{y}_i\}_{i=1}^{N}$ drawn from the joint distribution $p(\boldsymbol{x},\boldsymbol{y})$.

The MI contrastive log-ratio upper bound (CLUB) estimator [8] is based on a parameterized distribution $q_\theta(\boldsymbol{y}|\boldsymbol{x})$:

$$\mathcal{I}(\boldsymbol{x};\boldsymbol{y}) \leq \mathbb{E}[\frac{1}{N}\sum_{i=1}^{N}[\log p(\boldsymbol{x}_i|\boldsymbol{y}_i) - \frac{1}{N}\sum_{j=1}^{N}\log p(\boldsymbol{x}_j|\boldsymbol{y}_i)]]. \tag{7}$$

All the MI lower bounds require learning of a value function $f(\boldsymbol{x},\boldsymbol{y})$; the CLUB upper bound requires learning of a network approximation $q_\theta(\boldsymbol{y}|\boldsymbol{x})$. To make fair comparison, we set the value function and the neural approximation with one hidden layer and the same hidden units. For the multivariate Gaussian setup, the number of hidden units is 20. On the top of hidden layer outputs, we add the ReLU activation function. The learning rate for all estimators is set to $1 \times 10^{-4}$.

