# OpenReview forum: "Estimating Total Correlation with Mutual Information Bounds"
_NeurIPS.cc/2020/Workshop/DL-IG — NeurIPSW 2020: DL-IG Poster_

### Official Review · AnonReviewer1 · 2020-10-30
**Using recent mutual information estimators to estimate total correlation**

**Rating:** 8
**Confidence:** 5

**Review:**

A very nice workshop submission that shows how one can leverage recent advances in mutual information estimation in order to estimate the total correlation.

Once the total correlation is resolved recursively as a set of mutual informations, simply using recent MI estimators like InfoNCE, MINE, NWJ or CLUB can yield an estimate of the total correlation.  This is demonstrated on a simple joint Gaussian problem for which the Total correlation is known exactly.

I have some technical quals with the CLUB estimator (I don't believe it's a valid bound at all), but this doesn't directly affect this work, though the language might be tweaked to only refer to it as an estimator rather than a bound.

Very nice paper, short and sweet, a nice idea and nice simple demonstration it can work.

---

### Decision · Program_Chairs · 2020-11-07

Accept (Poster)